# Atypical Viral Infections in Gastroenterology

**DOI:** 10.3390/diseases10040087

**Published:** 2022-10-15

**Authors:** Neira Crnčević, Zijah Rifatbegović, Mirsada Hukić, Sara Deumić, Emina Pramenković, Amir Selimagić, Ismet Gavrankapetanović, Monia Avdić

**Affiliations:** 1Department of Genetics and Bioengineering, International Burch University, Francuske revolucije bb, 71210 Ilidža, Bosnia and Herzegovina; 2Department of Abdominal Surgery, Clinic for Surgery, University Clinical Centre Tuzla, 75000 Tuzla, Bosnia and Herzegovina; 3Center for Disease Control and Geohealth Studies, Academy of Sciences and Arts of Bosnia and Herzegovina, Bistrik 7, 71000 Sarajevo, Bosnia and Herzegovina; 4Institute for Biomedical Diagnostics and Research Nalaz, Čekaluša 69, 71000 Sarajevo, Bosnia and Herzegovina; 5Department of Gastroenterohepatology, General Hospital “Prim. dr. Abdulah Nakas”, 71000 Sarajevo, Bosnia and Herzegovina; 6Clinic of Orthopedics and Traumatology, University Clinical Center Sarajevo, Bolnička 25, 71000 Sarajevo, Bosnia and Herzegovina

**Keywords:** enteric viruses, SARS-CoV-2, hantavirus, dysbiosis, microbiome, gastrointestinal tract

## Abstract

Enteric viruses are commonly found obligate parasites in the gastrointestinal (GI) tract. These viruses usually follow a fecal-oral route of transmission and are characterized by their extraordinary stability as well as resistance in high-stress environments. Most of them cause similar symptoms including vomiting, diarrhea, and abdominal pain. In order to come in contract with mucosal surfaces, these viruses need to pass the three main lines of defense: mucus layer, innate immune defenses, and adaptive immune defenses. The following atypical gastrointestinal infections are discussed: SARS-CoV2, hantavirus, herpes simplex virus I, cytomegalovirus, and calicivirus. Dysbiosis represents any modification to the makeup of resident commensal communities from those found in healthy individuals and can cause a patient to become more susceptible to bacterial and viral infections. The interaction between bacteria, viruses, and host physiology is still not completely understood. However, with growing research on viral infections, dysbiosis, and new methods of detection, we are getting closer to understanding the nature of these viruses, their typical and atypical characteristics, long-term effects, and mechanisms of action in different organ systems.

## 1. Introduction

In recent years, the morbidity rates for waterborne diseases have increased drastically and they have now become a public concern around the world [1,2]. Based on data published by the World Health Organization (WHO) in 2008, it is estimated that around 2.2 million people die yearly due to a lack of clean water, hygiene, and sanitation. Apart from this, millions of others experience non-fatal diarrhea [3].

These symptoms are caused by enteric viruses, representing commonly found obligate parasites in the gastrointestinal (GI) tract. When it comes to gastroenteritis or viral hepatitis, patients can extract between 10^5^ and 10^11^ viral particles per gram of stool. These samples are composed of many genera, including noroviruses, adenoviruses, parvoviruses, astroviruses, rotaviruses, Hepatitis E viruses, Hepatitis A viruses, and as enteroviruses that include polioviruses, Coxsackie viruses, and echoviruses [4,5]. Apart from being able to cause many non-bacterial infections in the GI tract, these viruses can also cause encephalitis, respiratory infections, paralysis, and meningitis. This is especially seen in immunocompromised patients with high mortality rates [6].

The infection with an enteric virus follows a fecal-oral route of transmission. The virus then infects the GI tract and directly interacts with the host microbiome [7]. One of the most interesting characteristics of these viruses is how easily they can be transferred between hosts. In most cases, less than 20 viral particles are needed to cause illness [8]. Furthermore, these viruses exhibit high resistance and stability to environmental stresses [9].

Humans can come in contact with enteric viruses in different ways. In many instances, enteric virus infections come from contaminated water and food, usually when sewage effluent water is involved. This is especially seen in developed countries, where food crops are sometimes grown on land irrigated with sewage-polluted water [6,10]. If animals are infected by an enteric virus, they are usually asymptomatic. This is especially seen in swine and cattle. However, in some cases, the viral infection can cause neurological disorders, abortion, and even mortality [11,12,13,14].

In recent years, scientists have deeply analyzed the interplay between enteric viruses, the host immune system, and the host microbiome [7]. In studies on mice published in 2011, the role of commensal bacteria in enteric viral infections was examined. These studies included reoviruses, polioviruses, and mouse mammary tumor viruses. To deplete the levels of commensal bacteria, the mice were given a mixture of oral antibiotics. As a consequence of this, the poliovirus infection in these mice was substantially attenuated when compared to their negative controls that possessed a normal microbiota. The antibiotic-treated mice experienced reduced mortality rates and an attenuated fecal shedding of the virus due to the reduced rate of intestinal viral replication. When the intestinal microbiota in the antibiotic-treated mice was rebuilt, the poliovirus pathogenesis was restored. These results highlight the effect of the host microbiome on poliovirus infection in the GI tract [15,16].

We still have a lot to understand about how bacteria, viruses, and host physiology interact. Despite this, a growing corpus of research is starting to illuminate the exciting role that commensal and probiotic microbes play in the host’s ability to defend itself against viral infections [17]. A crucial defense against bacteria and viruses that cause disease is the mucosal epithelium. New studies have shown that the microbiome impacts the mucus layer’s development and function. For instance, in the GI tract, the microbiota stimulates the expression of the genes that code for mucin 2, the primary component of the glycoprotein network that makes up GI mucus [18]. It is interesting to note that swine stomach mucins have been proven to protect epithelial cells from infection by several viruses, including a strain of influenza A virus and the human papillomavirus type 16 and Merkel cell polyomavirus [19]. Additionally, it appears that the cervicovaginal mucus is influenced by the nature of the vaginal microbiota. The diffusion of HIV-1 virions has been demonstrated to be inhibited by a *Lactobacillus crispatus*-dominant microbiota, as opposed to the rapid distribution seen with a *Lactobacillus iners*-dominant microbiota or when *Gardnerella vaginalis* is abundant [20].

Numerous disease models have been used to conduct substantial research on the effect of the gut microbiota on GI barrier permeability, particularly its relationship with hyperpermeability, sometimes known as the “leaky gut” [21]. Although it can be challenging to apply the findings of in vitro and animal studies to human populations, numerous studies indicate that probiotics can enhance mucosal barrier function [22].

Antimicrobial peptides are protective substances found in almost all living things, from bacteria to people [23]. All important bacterial lineages produce bacterial AMPs, sometimes referred to as bacteriocins, which have long been regarded as essential probiotic characteristics. Although some bacteriocins may potentially have antiviral effects, their route of action has received much less study than their antimicrobial effectiveness against bacterial infections. The information at hand suggests two different forms of action. One way to look at it is that some bacteriocins appear to have antiviral properties before viral entrance into human cells. In this regard, it has been discovered that duramycin, a class-1 bacteriocin made by *Streptomycetes*, blocks the co-receptor of the Zika virus, TIM1 [24].

On the other hand, some bacteriocins interfere with the late stages of the viral cycle to diminish cytopathic effects and viral release yield even though they do not impede viral entrance. For instance, a *Lactobacillus delbrueckii* 5 kDa bacteriocin has no impact on the early stages of influenza virus infection, but it does decrease the generation of viral proteins in infected cells [25]. Similar to this, HSV types 1 and 2 disrupt late infectious stages when exposed to the bacteriocin subtilosin, which is generated by species of the *Bacillus* genus. In particular, viral glycoprotein gD’s intracellular location is changed yet virus multiplication up to the protein synthesis step is unaffected. As opposed to subtilosin, enterocin CRL35 from *Enterococcus faecium* was discovered to reduce glycoprotein gD synthesis. This suggests that enterocin CRL35 disrupts the infectious cycle of HSV at a significantly earlier stage [26]. The aim of this paper was to highlight the atypical cases of viral infections in gastroenterology, as well as the underlining mechanisms behind them.

## 2. The Microbiome, Dysbiosis, and Viral Infections

Any modification to the makeup of resident commensal communities from those found in healthy individuals has been generically referred to as dysbiosis [27]. The symbiosis between the host and the bacteria is thought to be disrupted by this change, which could have adverse effects. A working group of the International Life Sciences Institute North America has lately tackled the issue of what a healthy human gut microbiota entails because it is a complex matter. The high level of intra- and interindividual variation in human microbiomes, the absence of validated biomarkers to define and measure microbiome–host interactions, and the fact that it is still unclear whether dysbiosis is the result of changes in human physiology and disease are some of the difficulties [28]. Dysbiosis can also be defined in terms of its functional aspects, such as the creation of harmful chemicals originating from microbes [29]. In Figure 1, the pathophysiology of gut dysbiosis is shown.

## 3. Detection of Enteric Viruses

There is a desperate need to detect viruses not only in humans but in water environments as well. Novel methods of detection that are reliable, cost-effective, and time-efficient could improve the overall quality of life of humans and decrease the mortality rates caused by enteric viruses worldwide. One of the significant challenges in water control represents determining the number of enteric viruses in water that can cause infections. Firstly, the concentration of enteric viruses in these water environments is deficient [30]. Secondly, it is challenging to separate infectious and non-infectious virions during the analysis, and lastly, there are many inhibitors in the processes of amplification and genome extraction [31,32,33].

All methods for analyzing water samples can be divided into two main parts, including sample concentration determination and virus detection. In the past, the most common methods used for purifying water from viruses included ultracentrifugation, entrapment ultrafiltration, electropositive membrane, adsorption/elution protocols, and others [34]. Nowadays, modern analytical detection methods can be applied in the purification processes. Techniques such as nucleic acid sequence-based amplification (NASBA), real-time NASBA, polymerase chain reaction (PCR), reverse-transcription PCR (RT-PCR), and enzyme-linked immunosorbent assay (ELISA) have become more popular [35].

## 4. Atypical Viral Infections in Gastroenterology

### 4.1. SARS-CoV-2

Today, the COVID-19 outbreak represents one of the most significant worldwide threats to public health. It is caused by severe acute respiratory syndrome coronavirus 2 (SARS-CoV-2), first identified in a patient in Wuhan City, Hubei Province, China [36]. This patient had an unusual case of pneumonia. The first cases appeared in early December 2019. By the end of the month, the World Health Organization (WHO) regional office in Beijing noticed several other cases in Wuhan City that had similar symptoms [37].

The high-risk groups of this infection were noticed to be older male patients who have cardiovascular diseases, hypertension, or diabetes [38]. The infection occurs during breathing or in direct contact with droplets and aerosols that contain the virus and usually happens when the infected person coughs or sneezes. The virus in the nasal cavity then goes into the epithelial cells of the host. This entry into the epithelial cells is possible because of angiotensin-converting enzyme 2 (ACE2) receptors. These receptors can commonly be found on epithelial cells that line the digestive tract and the respiratory system [39,40].

Since the apical sides of lung epithelial cells in the alveolar space have a high expression of the ACE2 receptor, SARS-CoV-2 can quickly enter and destroy these cells. This is in concordance with the fact that many early cases of lung damage caused by these infections can be seen in the distal airways. When it comes to the innate immune response in the airway, the primary cells involved include epithelial cells, dendritic cells, and alveolar macrophages, that fight viruses before adaptive immunity [41].

Interestingly, young children and infants are usually at high risk for hospital admissions because of potential respiratory infections. However, in COVID-19 patients, recent data shows that pediatric patients have generally milder symptoms when compared to older patients. There is a strong correlation between the severity of COVID-19 and the amount of viral loads. Since children can have less viral load in these infections, this could explain the difference in severity [42]. Many hypotheses try to explain this. The first hypothesis suggests a difference in expression levels of the ACE2 receptor in these two age groups. The ACE2 receptor is usually more abundantly expressed in well-differentiated ciliated epithelial cells. This could explain why pediatric cases exhibit milder symptoms [43].

Moreover, the ACE2 expression could explain differences in disease severity among different gender groups. ACE2 gene can be found on the X-chromosome. Men have higher circulating ACE2 levels than women [44]. This could explain the differences in disease severity and mortality between men and women [45,46].

More specifically, immunofluorescent techniques show that the ACE2 receptor is abundantly expressed in glandular cells of the rectal, duodenal, and gastric epithelia. On the other hand, the esophageal epithelium composed of squamous cells has fewer expression levels of the ACE2 receptor when compared to glandular cells [47]. Additionally, the ACE2 receptor is expressed in cholangiocytes and hepatocytes, which could explain a potential association between COVID-19 infection and liver complications [48].

The symptoms of COVID-19 can range from mild to severe illness and even, in some cases, death [49]. Even though the most common symptoms of this virus include shortness of breath, fever, and a dry cough, there have also been cases in which patients experience gastrointestinal symptoms [50]. For COVID-19, gastrointestinal symptom manifestations are approximately 11% to 53%. Additionally, almost half of the patients experienced at least one symptom such as diarrhea, abdominal pain, nausea, and vomiting [51]. In severe cases, the infection can lead to multiple organ failures, especially in respiratory cases [52].

A meta-analysis involving 47 studies with 10,890 patients in total suggested that the most common gastrointestinal manifestations of this virus include vomiting and nausea (7.8% of patients), diarrhea (7.7% of patients), and abdominal pain (2.7% of patients) [48]. In a study by Ferm et al., they found that out of 892 patients in total, the following gastrointestinal symptoms were noticed: loss of taste, nausea, vomiting, abdominal pain, diarrhea, and loss of appetite. Out of these, diarrhea and nausea had the most significant percentage of patients, at 19.8% and 16.6%, respectively [53].

Human lungs and gut possess distinct niches of microorganisms. Both the lungs and the gut contain *Bacteroidetes* as well as *Firmicutes* as their main bacteria. In the lungs, *Proteobacteria* can also be one of the most prevalent microorganisms [54].

The gut microbiota can affect pulmonary health in many different ways. The relationship between these two environments is seen through the so-called “gut–lung axis”, which enables them to communicate with each other [55]. This communication happens in both ways. Endotoxins can affect the lungs through blood. The newly formed inflammation in the lungs can then impact the gut microbiota [56].

Bacterial infections often occur as a complication due to viral pneumonia. This can heavily affect the course of the disease, which is especially dangerous for critically ill patients [57]. However, when it comes to viral pneumonia, there is not enough information about bacterial coinfections that are a result of COVID-19. Since the COVID-19 pandemic occurred just recently, there are not many studies that deny nor confirm this. In most articles about COVID-19, only a tiny number talk about secondary infections [58].

In recent studies, the association between microbiome and the severity of SARS-CoV-2 has been mentioned. In these studies, the fecal microbiome among 7 antibiotic-free patients had experienced other rigors of infection, including mild, moderate, and severe. Association with COVID-19 severity was observed with 23 bacteria taxa. Out of these, 15 belonged to the phylum *Firmicutes*. More specifically, seven of these were negatively correlated with the development of the disease. In the *Firmicutes* phylum, *C. hathewayi*, *Coprobacillus*, and *Clostridium ramosum* were the main bacteria that positively correlated with the severity of the illness [59].

Gut microbiome dysbiosis in SARS-CoV-2 was also analyzed in a study from 2020. In COVID-19 patients, the abundance of bacteria that produce butyrates such as *Eubacterium rectale, Faecalibacterium Prausnitzii, Clostridium leptum*, and *Clostridium butyricum* was observed. However, when compared to their negative controls, these patients also possessed an increased amount of common opportunistic pathogens such as *Enterococcus* and *Enterobacteriaceae* [60].

Similar studies focused on the specific gut microbiota fluctuations in COVID-19 patients were also conducted. It was found that the feces of COVID-19 patients were enriched with opportunistic pathogenic genera, including *Veillonella, Streptococcus, Rothia*, and *Actinomyces*, when compared to their negative controls. In the feces of negative controls, an abundance of genera *Fusicatenibacter, Faecalibacterium*, and *Romboutsia* were also noticed [61].

Even though COVID-19 can be airborne in most cases, some believe the virus could also be transmitted following an oral-fecal route. There is concern that the gastrointestinal tract could be a repository for reinfections. This is because the virus was found in the surface cells of the colon, small intestine, and colon. The number of atypical cases and clinical symptoms involving the gastrointestinal tract is increasing. Additionally, infected individuals were shown to have a prolonged shedding of viral particles in their stools [47,52,62,63,64].

Even though the oral-fecal transmission explanation has recently gained much attraction, the origin of COVID-19 in the gastrointestinal tract and its clinical shedding in patients’ stools is still controversial. The reverse transcription-polymerase chain reaction could detect only the viral fragments as opposed to the whole virus. Additionally, stool cultures for this virus cannot be used to confirm its viral presence accurately. More studies are needed to consider the incubation period, transmission mechanism, and duration of infectivity [65].

More severe gastrointestinal manifestations of COVID-19 include bloody diarrhea. In rare cases, patients can also experience hemorrhagic colitis and constipation [63,66]. In a study conducted by Lin et al., COVID endoscopic evaluation was performed for COVID-19 patients with gastrointestinal bleeding showed a large number of herpetic erosions and ulcers [67]. Additionally, in a study by Seeliger et al., COVID-19 patients with severe symptoms were shown to have ischemic and ulcerative changes using rectosigmoidoscopy [68].

Viral load can be measured using real time-PCR. An increasing viral load implies that there is an increase in the patient’s viral particles which leads to more severe symptoms. On the other hand, a lower or down-trending viral load could point to milder symptoms or recovery stages. For respiratory tract samples, the viral load usually reaches a peak around the onset of symptoms. It then gradually decreases in the next 1–3 weeks. In most cases, RNA generally becomes undetectable after two weeks of symptom onset [69].

For stool samples, viral shedding can be present even after the respiratory clearance. However, it is usually erratic. In some cases, it can even continue for approximately two months after the symptom onset. The viral load needs to be examined in different organs to form an accurate association between viral load, disease progression, and symptom severity [70].

### 4.2. Hantavirus

Hantavirus belongs to the family *Bunyaviridae* [71]. They possess three negative-sensed RNA segments including small, medium, and large segments. These segments are responsible for coding RNA-dependent RNA polymerase, surface envelope proteins G1 and G2, and nucleocapsid protein. These viruses also possess a spherical lipid envelope [72].

This is a rare virus that can attack humans and causes human pulmonary syndrome (HPS) and hemorrhagic fever with renal syndrome (HFRS). Information about the symptoms that possibly occurred due to this virus date back to the first millennium and the Middle Age in England and China [73,74]. The mortality rate of hemorrhagic fever with renal syndrome and HPS is around 12% and 40%, respectively [75]. The risk groups are immunocompromised patients [76].

This virus can be spread through rodents. A person can get infected if they inhale aerosols contaminated with the virus in the form of rodent excreta [77]. Because of this, some studies suggest that people who live near infected rodents have a higher chance of getting infected [78,79].

HCPS can lead to cardiovascular irregularities and pneumonia. HFRS is characterized by hemorrhagic manifestations and renal failure. The hemorrhagic manifestations can range from petechiae to intense internal bleeding [80]. The incubation period for HFRS is usually between 10 days to 6 weeks. This is then followed by a febrile phase in which nonspecific symptoms such as abdominal pain, myalgia, and neurological, gastrointestinal, and cardiovascular symptoms appear [81]. It can lead to dangerous complications such as mucosal bleeding, petechiae, and epistaxis. In the later stages of the disease, pleural and gastrointestinal bleeding as well as hematuria can occur [82]. In rare cases, this can also lead to spleen hemorrhage [83] and even panhypopituitarism [84].

### 4.3. Herpes Simplex Virus

Herpes simplex virus belongs to the genus *Simplexvirus* subfamily *Alphaherpesvirinae* in the family *Herpesviridae* [85]. There are four main parts of a mature herpes simplex virus, including an outer membrane envelope, a core that contains the viral DNA, an icosahedral capsid as well as the tegument. The outer membrane envelope possesses many glycoprotein spikes, and the tegument has specific additional viral proteins needed for the proper functioning of the virus. Moreover, the virus contains a double-stranded DNA viral genome. It is secure inside the viral capsid shell [86,87,88,89].

This virus can be transmitted if a person comes in contact with contaminated bodily fluids such as saliva. The early beginnings of the infection are characterized by its replication at the site of the infection. The virus then travels to the dorsal root ganglia down an axon. This is the location where the virus achieves its latency. In this period, the virus is in a non-infectious state. At different periods, HSV-1 can reactivate itself and attack the host [90,91,92,93]. As a result of this, vesicular eruptions can appear usually in the orobial mucosa. Apart from orobial herpes, other presentations such as herpes gladiatorum, ocular herpes infection, herpes encephalitis, herpes sycosis, and herpetic whitlow can occur [94].

Although herpetic infection can affect any part of the GI tract, it most frequently affects the esophagus and anorectum [95]. Although immunocompromised people frequently experience herpes infections of the gut, this group is not the only one [96]. Herpetic esophagitis patients also have GI hemorrhage, dysphagia, chest pain, nausea, and vomiting. At the time of diagnosis, a lot of people had herpes infection that had spread. The most typical cause of nongonococcal proctitis in homosexual males is herpetic proctitis. Severe anorectal pain, tenesmus, constipation, discharge, hematochezia, and fever are typical symptoms of patients. Along with inguinal lymphadenopathy, concomitant neurologic symptoms such trouble urinating and paresthesias in the buttocks and upper thighs are extensively characterized [97]. The most frequent gross esophageal finding is an ulcer, which is frequently accompanied by an exudate. However, a nonspecific erosive esophagitis affects many people. Perianal vesicles are frequently present in herpetic proctitis. Mucosal friability and ulceration are examples of cytological findings. Vesicles can occasionally be detected in the anal canal or rectum [98].

No matter the location, localized ulceration, neutrophils in the lamina propria, and an inflammatory exudate that frequently incorporates shed epithelial cells are typical histologic findings. Additionally, crypt abscesses and perivascular lymphocytic cuffing may be detected in the anorectum. Only a small percentage of biopsy specimens include multinucleate large cells and distinctive viral inclusions. The squamous epithelium near ulcer margins and sloughed cells in the exudate are the ideal places to look for viral inclusions. Among all diagnostic tools, viral culture is the most useful. Specific techniques include in situ hybridization and immunohistochemistry. Other viral infections that might affect the GI tract, such as CMV and varicella-zoster, are primarily included in the differential diagnosis. When herpetic infection is present, mixed infections are frequently seen. Herpetic infections frequently self-limit in immunocompetent patients, but they can spread and cause life-threatening sickness in immunocompromised people [99].

### 4.4. Cytomegalovirus

Immunocompetent and immunocompromised individuals can contract cytomegalovirus (CMV) infection anywhere in the GI tract. In individuals with a compromised immune system, such as those with AIDS, and following solid organ or bone marrow transplantation, CMV is best known as an opportunistic virus. In healthy individuals, first infections typically resolve on their own. The patient’s immune system and the infection site influence the symptoms. The most typical clinical signs are weight loss, fever, stomach pain, and diarrhea (either bloody or watery). Hypertrophic gastropathy and protein-losing enteropathy, similar to Ménétrier’s disease, are rare but significant conditions connected to pediatric CMV infection [100].

Additionally, chronic GI conditions such as Crohn’s disease and ulcerative colitis may have secondary CMV added. In these circumstances, CMV superinfection is linked to steroid-refractory sickness, toxic megacolon, and a greater mortality rate in addition to exacerbations of the underlying illness. Immunohistochemistry testing for CMV is advised by certain experts as part of the routine analysis of biopsies in patients with steroid-refractory ulcerative colitis [101].

A stunning array of gross lesions are brought on by CMV. The most typical type is ulceration. There may be one or more ulcers, which may be shallow or deep. Linear ulcers and segmental ulcerative lesions might resemble Crohn’s disease. Mucosal bleeding, pseudomembranes, and obstructive inflammatory masses are some other gross abnormalities. The CMV infection has a broad histologic spectrum, ranging from hardly noticeable inflammation to profound ulcers with apparent granulation tissue and necrosis. Routine H&E preparations may reveal distinctive “owl’s eye” viral inclusions, which can either be intracytoplasmic or intranuclear [100]. It is more common to find inclusions in stromal and endothelial cells than in epithelial cells. CMV inclusions are frequently detected deep in ulcer bases as opposed to the borders of ulcers or the surface mucosa, unlike adenovirus and herpes. Adjacent nuclei might be more prominent, look smeared, or look like ground glass, but they do not have the usual inclusions. Cryptitis, a mixed inflammatory infiltrate that often contains many neutrophils, and mucosal ulceration, is associated with histologic findings [102].

It is possible to notice multiple apoptotic enterocytes and crypt abscesses, crypt atrophy, and crypt loss. Immunocompromised patients may develop distinctive inclusions with essentially no concomitant inflammatory reaction. When only a few unique inclusions are observed in biopsy specimens, the diagnosis could be easily overlooked. Immunohistochemistry and multilevel examinations may be extremely helpful in locating the seldom cells that have inclusions. Viral culture, PCR assays, in situ hybridization, serologic analyses, and antigen tests are other diagnostic tools. However, the presence of CMV in culture does not necessarily indicate a current infection since the virus might be excreted for months to years following primary infection [100].

Other viral infections, particularly adenoviruses, are primarily included in the differential diagnosis of CMV. Adenovirus inclusions are often intranuclear, crescent-shaped, and typically seen in the surface epithelium. CMV inclusions naturally occur in stromal or endothelial cells, have an “owl’s eye” shape, and can be seen in the nucleus or cytoplasm. The most striking similarity between CMV and adenovirus infection occurs during the ballooning degeneration stage, right before cell lysis [103].

Given the similarity of the clinical and histologic characteristics, distinguishing between CMV infection and graft-versus-host disease in recipients of bone marrow transplants may be particularly challenging. In this situation, it is best to rule out CMV infection using immunohistochemistry or in situ hybridization investigations because delaying antiviral medication if CMV infection is not detected. Additionally, various ailments might coexist. When there is a lot of apoptosis, crypt necrosis, and dropout, and there is not much inflammation, graft-versus-host disease is more likely to occur. Graft-versus-host disease is made more likely by the presence of active nests of endocrine cells [102].

### 4.5. Calicivirus

Caliciviridae represents a family of positive-sense single-stranded RNA viruses that infect both humans and animals. Phylogenetically speaking, these viruses can be divided into the following genera: Lagoviruses, Vesiviruses, Sapoviruses, and Noroviruses [104]. Sapoviruses and Noroviruses are the most common genera that cause acute gastroenteritis [105,106].

In recent decades, an increased prevalence of gastroenteritis has been noticed worldwide. A higher mortality rate has also been seen in developing countries. Apart from this, gastroenteritis outbreaks can occur in institutions such as hospitals [106,107,108,109]. A large number of these outbreaks have occurred due to contaminated food [107].

Noroviruses follow an oral-fecal transmission route, but a few airborne transmission cases have also been reported. The infection dose of these viruses is very low, and the incubation period is from 10 to 51 h. One-third of infected individuals shed the virus before the symptoms occur [110].

Human norovirus is sometimes referred to as “stomach flu” or “winter vomiting disease”. From 2006 to 2010, CDC reported that more than half of outbreaks caused by foodborne diseases were associated with this virus. During this period, the European Union also reported a large number of infections caused by this virus [111].

The most common symptom in adults is vomiting [112]. A large number of patients also sometimes have only short-term and mild symptoms that can be suppressed by oral hydration, rest, and intravenous replacement of electrolytes. However, complications can arise in the elderly and infants since they are more sensitive to volume depletion. Additionally, complications are also seen in immunocompromised patients who have received an organ transplant. This is especially seen in infants with intestinal transplants. They often experience symptoms such as persistent diarrhea. If this occurs, their immunosuppressive therapy is usually reduced [113].

Sapovirus infection outbreaks have also been more frequent in recent years [114]. When compared to noroviruses, sapoviruses cause milder symptoms, but like noroviruses, immunocompromised patients, infants, and the elderly are more prone to hospitalization [115,116]. On top of that, there are also differences in their most common symptoms. For sapoviruses, the most common symptom is diarrhea as opposed to noroviruses which usually cause vomiting, and low-grade fever [117]. In Table 1, the comparison between the mentioned viruses is shown.

## 5. Conclusions

Enteric viruses are stable and unusually resistant obligate parasites in the gastrointestinal tract that cause symptoms such as vomiting, abdominal pain, and diarrhea. The mechanisms of action and long-term effects of viral infections on host physiology are still not completely understood. This data gap is especially true for atypical viral infections, so more studies on this topic need to be conducted. The data obtained from these studies could be used to find the most optimal method of detection and treatment option. The viruses mentioned in this paper include SARS-CoV2, hantavirus, herpes simplex virus, cytomegalovirus, and calicivirus. These viruses share some commonly exhibited symptoms. However, their atypical gastrointestinal manifestations differ drastically. Many believe that the association between dysbiosis, any modification to the makeup of resident commensal communities from those found in healthy individuals, and viral infections are crucial in understanding susceptibility to these viruses and the treatment outcome. By understanding the underlining mechanisms of dysbiosis, we are getting closer to understanding atypical viral infections that cause gastrointestinal symptoms.

## Figures and Tables

**Figure 1 diseases-10-00087-f001:**
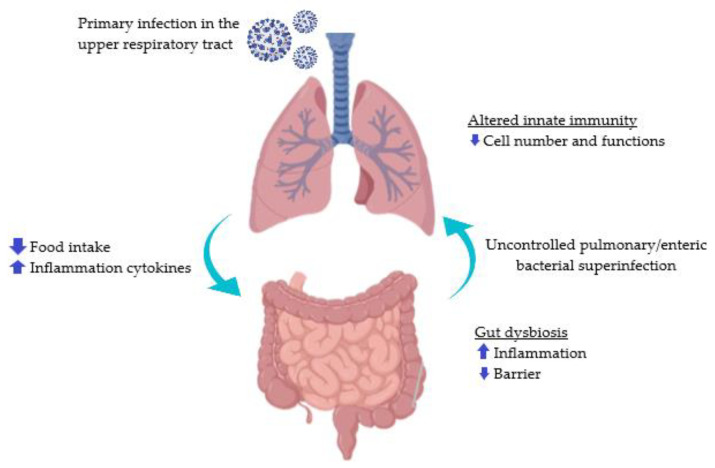
Pathophysiology of gut dysbiosis. Data taken and adapted from [27,28,29].

**Table 1 diseases-10-00087-t001:** Comparison of virus characteristics. Data from [38,50,51,76,81,82,83,84,94,96,97,98,99,100,106,107,108,109,114].

Atypical Viral Infection	Genetic Material	Symptoms		Risk Groups
		Common	Rare	
SARS-CoV-2	RNA	Shortness of breath [50] Fever [50] Dry cough [50]	Diarrhea [51] Abdominal pain [51] Nausea [51] Vomiting [51]	Older males with cardiovascular diseases, hypertension, or diabetes [38]
Hantavirus	RNA	Abdominal pain [81] Myalgia [81] Neurological [81]	Mucosal bleeding [82] Epistaxis [82] Spleen hemorrhage [83] Panhypopituitarism [84]	Immunocompromised patients [76]
HSV	DNA	Herpes [94] Ulcer [98] Chest pain [97] Fever [97]	Multinucleated large cells [99] Distinctive viral inclusions [99]	Immunocompromised patients [96]
CMV	DNA	Weight loss [100] Fever [100] Stomach pain [100] Diarrhea [100]	Hypertrophic gastropathy [100] Protein-losing enteropathy [100]	Immunocompromised patients [100]
Calicivirus	RNA	Diarrhea [114] Vomiting [114] Gastroenteritis [106,107,108,109] Low-grade fever [114]	/	Infants [114] Immunocompromised patients [114] Elderly patients [114]

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
