# Peer review of "Atypical Viral Infections in Gastroenterology"

_diseases, 2022, doi:10.3390/diseases10040087_

Round 1

Reviewer 1 Report

I read with great pleasure the manuscript entitled Atypical Viral Infections in Gastroenterology”

First of all, the manuscript should be thoroughly revised regarding typographical and grammar errors.

A lot of data is presented but in a chaotic way. The data should be presented systematically to determine who is prone to this type of infection (children, immuno-compromised/competent…?) It should be apparent are the authors analyzing children or adults. What is the clinical presentation of the infection? What type of pathological lesion does the infection cause? And if the topic is atypical viral infections in Gastroenterology, liver infections should be mentioned as well.

Why a chapter is dedicated to typical viral infections if the topic is atypical viral infections? Mentioned HSV and CMV, are atypical viral infections for GI tract…So this should also be corrected.

In my opinion, discussing probiotics (chapter 6) is unnecessary, especially mentioning treatment options.

Also, chapter 7 is out of the topic. Gut microbiota should be mentioned and discussed in the beginning for explanation of susceptibility to atypical viral infections.

In the conclusion, it is not clear whether are we talking about atypical viral infections or atypical clinical symptoms. The authors are mixing typical and atypical strains. In the end, I could not realize the aim of this review-discussing dysbiosis or the presentation of atypical viral infections. Despite the importance of SARS-CoV2 nowadays,

In the era of everyday advances in immunomodulatory and immunosuppressive therapy, conditions which immunocompromise, and of course still present pandemic od SARS-CoV2 infection, drawing attention to this type of infection is very important. A more correct explanation of the pathophysiological mechanisms and clinical presentation of the disease in the digestive tract is of great importance. Therefore, for this manuscript to be worth reading, a significant correction is necessary.

Reviewer 2 Report

The authors of this manuscript reviewed about atypical viral infections in gastroenterology. I think the topic of this manuscript is interesting. However, this manuscript still have several flaws in its content which need revision to improve the quality. My comments regarding this manuscript can be seen below:

1) Introduction: I do not see sufficient background or reason in the Introduction section to perform this review. The authors should explained more why their review about atypical viral infection in gastroenteritis is important. What is lacking or still not clear from the current evidences? Is there literature gap? These should be highlighted before the statement about purpose of this study.

2) In the typical viral infections in gastroenterology, why do the authors choose herpes simplex virus and cytomegalovirus instead of norovirus, astrovirus, and other more common viral infection? Moreover, why do the authors put rotavirus under adenovirus? Rotavirus and adenovirus is two different viruses. Please explain.

3) Since the title of this study is "Atypical viral infections in gastroenterology", why are there only two viruses explained in this study (SARS-CoV-2 and Hantavirus)? How about other atypical viral infections which may also cause gastrointestinal symptoms, such as Calicivirus, Monkeypox, MERS-CoV, etc?

4) In the description about SARS-CoV-2, the authors have mentioned about high-risk group which includes older age, patients with hypertension, diabetes, and cardiovascular disease. The authors may also look into these studies to enrich the evidences in their review (https://doi.org/10.1016/j.archger.2020.104299, https://doi.org/10.1016/j.diabres.2021.109031)

5) Throughout this manuscript, I do not see any Tables or Figures. I think this manuscript will benefit from addition of Tables and Figures.

- One table may explain the atypical viral infections in gastroenterology along with their manifestation and if there's any differences from the typical viral infections

- One more table may explain the results from various clinical trials regarding efficacy of probiotics for gastrointestinal disorders

- One Figure may explain about the pathophysiology of gut dysbiosis or microbiome in gastrointestinal disorders and the role of probiotics

Round 2

Reviewer 1 Report

After the suggested revision, the manuscript is adequate for publishing.

Reviewer 2 Report

Thank you for revising the manuscript according to my comments. The manuscript is much better now and can be accepted for publication.